# Palliative Care in Pediatric Pulmonology

**DOI:** 10.3390/children8090802

**Published:** 2021-09-13

**Authors:** Taylor Baumann, Shailendra Das, Jill Ann Jarrell, Yuriko Nakashima-Paniagua, Edith Adriana Benitez, Maria Carolina Gazzaneo, Natalie Villafranco

**Affiliations:** 1Department of Pediatrics, Texas Children’s Hospital, Baylor College of Medicine, Houston, TX 77030, USA; Taylor.Baumann@bcm.edu; 2Section of Pediatric Pulmonary Medicine, Department of Pediatrics, Texas Children’s Hospital, Baylor College of Medicine, Houston, TX 77030, USA; Shailendra.Das@bcm.edu (S.D.); Gazzaneo@bcm.edu (M.C.G.); 3Section of Palliative Care, Department of Pediatrics, Texas Children’s Hospital, Baylor College of Medicine, Houston, TX 77030, USA; Jill.Jarrell@bcm.edu; 4Section of Palliative Care, Department of Pediatrics, Hospital Civil de Guadalajara, Guadalajara 44280, Mexico; yuritita@yahoo.com (Y.N.-P.); ady_benitezv@hotmail.com (E.A.B.)

**Keywords:** pediatric pulmonology, palliative care, communication, psychosocial support, primary palliation, quality of life, cystic fibrosis, bronchopulmonary dysplasia, neuromuscular disease, neuromuscular disease, pulmonary hypertension, lung transplant

## Abstract

Children with End Stage Lung Disease (ESLD) are part of the growing population of individuals with life-limiting conditions of childhood. These patients present with a diverse set of pulmonary, cardiovascular, neuromuscular, and developmental conditions. This paper first examines five cases of children with cystic fibrosis, bronchopulmonary dysplasia, neuromuscular disease, pulmonary hypertension, and lung transplantation from Texas Children’s Hospital. We discuss the expected clinical course of each condition, then review the integration of primary and specialized palliative care into the management of each diagnosis. This paper then reviews the management of two children with end staged lung disease at Hospital Civil de Guadalajara, providing an additional perspective for approaching palliative care in low-income countries.

## 1. Introduction

Children with End Stage Lung Disease (ESLD) make up approximately 9% of the growing population of individuals with life-limiting conditions of childhood [1]. These patients present with a diverse set of pulmonary, cardiovascular, neuromuscular, and developmental conditions. This paper will examine five cases of children with ESLD at Texas Children’s Hospital with conditions including cystic fibrosis (CF), bronchopulmonary dysplasia (BPD), neuromuscular disease (NMD), pulmonary hypertension, and lung transplantation. We will review the expected clinical courses associated with each diagnosis and then discuss how palliative care principles can be integrated into treatment to achieve quality, goal-concordant care. We will then examine two cases of individuals with cystic fibrosis managed by multidisciplinary teams in Hospital Civil de Guadalajara to provide an additional perspective for approaching palliative care in low-income countries. This article should serve as a guide to primary physicians, pulmonologists, and specialized palliative care (SPC) providers to improve team-based care of patients with ESLD.

In ideal situations, primary care providers, pulmonology teams, and other multidisciplinary team members work together to provide primary palliation to patients with ESLD. This includes early use of palliative care as part of holistic chronic disease management, strengthening palliative care capabilities in primary care, and collaboration with SPC services where they exist [2]. Dedicated SPC teams should be consulted when advanced clinical treatments, complicated decision-making, and social or spiritual needs extend beyond what the primary team can provide. Primary care providers, pulmonary providers, psychiatry providers, psychology providers, and SPC providers should work together to address the palliative care principles of relieving suffering, improving the child’s quality of life, facilitating informed decision making, and coordinating of care among healthcare teams [3].

## 2. Cystic Fibrosis

Petre is a 17-year-old boy with advanced cystic fibrosis disease. He is on a regimen consisting of inhaled therapies, oral medications, and mechanical assisted airway clearance; he uses these therapies three times each day. Even so, Petre’s functional status has continued to decline, and his primary team predicts that Petre is likely to die within the next two years without a lung transplant. Recently, he was started on oxygen therapy by nasal cannula for daytime and nighttime use, as well as nutritional supplementation via gastrostomy tube overnight. Petre expresses frustration with the time investment required by his current therapies and concern over both the surgery and the lifelong medical therapy associated with a lung transplant. He asks for guidance with his therapeutic options moving forward.

### 2.1. Cystic Fibrosis Clinical Course

Cystic fibrosis is a life-limiting inheritable disease caused by a defect in the cystic fibrosis transmembrane conductance regulator (CFTR) protein, which ultimately impairs mucus hydration, mucociliary clearance, and microbial defense. This causes a multisystemic disease with potential consequences, including lung remodeling, recurrent infection, respiratory failure, pancreatic insufficiency, diabetes, hepatobiliary dysfunction, gastrointestinal obstruction, and reduced fertility. The degree of a patient’s symptoms is often a reflection of the type and zygosity of the mutation present in that individual. Although infants with cystic fibrosis rarely survived beyond early childhood when first described by Dr. Dorothy Hansine Andersen in 1938 [4], subsequent advances in antibiotics, protein modulation, supportive care, and genetic modifiers now provide many individuals diagnosed with cystic fibrosis with more opportunities to lead full lives.

For most patients with cystic fibrosis, the standard approach to treatment requires the patient and his or her family to commit to a time-consuming daily regimen which often includes pulmonary toilet, mucolytic agents, inhaled antibiotics, mealtime oral pancreatic enzymes, vitamin supplementation, oral protein modulators, and insulin for cystic fibrosis related diabetes [5]. As lung disease progresses over time and colonizing bacteria take advantage of the patient’s remodeled lungs, individuals can expect to suffer from intermittent episodes of bronchopneumonia which often require prolonged outpatient or inpatient antibiotics for management.

Due to the high morbidity and mortality associated with CFTR dysfunction, cystic fibrosis has historically been considered primarily as a disease of childhood. As recently as 2013, the expanding population of adults only accounted for half of the total cystic fibrosis population in the United States [6]. However, in large part due to novel medications targeting non-functional CFTR protein, individuals now born in the US can expect to live to be 48 or older [7]. Most recently, elexacaftor-tezacaftor-ivacaftor (Trikafta™) has revolutionized the treatment of cystic fibrosis, significantly reversing declining lung function, preventing disease exacerbations, and improving the quality of life for qualifying patients with at least one Phe508del mutation [8]. Although this genetic requirement applies to over 90% of individuals with cystic fibrosis in the United States, many patients are limited in their access to this medication due to their microbiology, adherence, uncontrolled diabetes, or alternate genetic mutations. The high cost of this medication represents an additional barrier to access, especially for patients outside of the United States. To further complicate matters, variations in the global distribution of CFTR mutations play an important role. For example, one study found only approximately 25% of the cystic fibrosis population in Turkey would qualify for the medication based on genetic factors alone [9].

### 2.2. Palliative Care in Cystic Fibrosis

Cystic fibrosis presents unique palliative care challenges due to the disease’s multi-systemic complications, complex treatment regimens, psychosocial impact, associated peer isolation, and unique relationship with lung transplantation [10]. As most individuals with cystic fibrosis rely on a team of healthcare providers to coordinate their management, the Cystic Fibrosis Foundation (CFF) recommends that each medical team member participates in providing primary palliative care to their patients. This incorporates the principles of managing symptom progression, reducing patient suffering, and respecting individual goals at the time of diagnosis and throughout the individual’s life. Providers should have a low threshold to contact SPC services when this management reaches beyond the scope or complexity of the primary team. In a recent 2020 study, a group of over 500 individuals, including people with cystic fibrosis, their caregivers, and their medical teams identified multiple key palliative care needs, including assistance with the complexity of treatment regimens, emotional support, mental health management, and guidance through shared decision making with advanced care planning [10]. In order to better identify these needs on an individual level, the CFF also recommends administering the standardized Integrative Palliative Care Outcome Scale starting at age 12, annually, and with disease milestones such as changes in severity or functional decline [11].

Among the increasingly complex treatment regimens involved with cystic fibrosis care, the effective pulmonary toilet remains a cornerstone of therapies that reduce the symptomatic progression of cystic fibrosis, slowing the decline of lung function over time. However, these benefits come at a significant cost of time, as many individuals with cystic fibrosis will need to engage in an average of 2–3 h of temporally spaced breathing treatments and chest physiotherapy each day [12,13]. Although critical in preventing functional decline, it is impossible to measure how the cumulative thousands of hours an individual with cystic fibrosis spends engaged in the pulmonary toilet might impact developmentally appropriate socialization, personal growth, academic performance, and eventually job opportunities. It is therefore important to balance the physical benefits with the developmental and psychosocial burden of these regimens to create the least restrictive regimen possible as a key step in improving a patient’s quality of life.

Growing evidence supports the connection between chronic illness, anxiety, and depression [14]. It is, therefore, unsurprising that the prevalence of these comorbidities is higher in individuals with cystic fibrosis (9–46%) and their caregivers (20–35%) compared to the general population [15]. These high rates of anxiety and depression underscore the importance of screening in both groups, with the Patient Health Questionnaire 9 Item Scale (PHQ-9) and Generalized Anxiety Disorder 7 Item Scale (GAD-7) offering quick, validated options for recommended annual assessment [11]. Appropriate management of mental health has clear consequences on illness progression, as anxiety and depression have been strongly associated with both treatment adherence and health-related quality of life [16,17].

Despite significant recent advances in cystic fibrosis care, the clinical course for most patients with cystic fibrosis is characterized by worsening lung function and eventual clinical decline. In a recent survey, adults with cystic fibrosis, their caregivers, and their team members all identified decision-making support and advanced care planning as two key palliative care needs [10]. Although the majority of cystic fibrosis providers report having discussions with their patients detailing palliative care management, most patients report a lack of understanding of their available options. Many adults with cystic fibrosis die in the ICU if palliative assistance is not expressly provided during advanced care planning. In response to this, the Cystic Fibrosis Foundation has encouraged providers to become increasingly proactive in involving Palliative Care in regular cystic fibrosis management. Previously, the Cystic Fibrosis Foundation recommended Palliative Care involvement after a patient reached severe stages of the disease or initiated evaluation for a lung transplant. These recommendations have since been updated, recommending earlier involvement of SPC services, during the moderate stages (FEV1 < 50%) of disease. Once initiated, SPC teams should remain involved throughout the course of an individual’s treatment, occurring alongside and in partnership with transplant teams [11].

## 3. Bronchopulmonary Dysplasia

Baby boy Sanna is a six-month-old infant, born at 25 weeks gestation, and diagnosed with bronchopulmonary dysplasia after failure to extubate by 36 weeks postmenstrual age. He has now failed three extubation attempts due to an inability to maintain ventilation on continuous positive airway pressure (CPAP). He is unable to participate fully in therapies due to the need for sedation to maintain a safe airway. Sanna’s parents have been counseled that their child requires a tracheostomy for long-term ventilation prior to discharge from this hospital. They are hesitant to move forward due to uncertainties with the tracheostomy procedure and with Sanna’s life at home on a ventilator. They ask if the team could continue serial attempts at extubation as an alternative.

### 3.1. Bronchopulmonary Dysplasia Clinical Course

Bronchopulmonary dysplasia is characterized by impaired pulmonary development with long-term effects on lung function that can last into adulthood [18]. The pathogenesis of BPD is complex and difficult to target therapeutically [19]. Definitions for BPD vary regionally and across institutions [20]. Currently, most institutions recognize BPD as a clinical diagnosis, contingent on the degree of an infant’s degree of respiratory support by 36 weeks postmenstrual age (PMA) [21,22,23,24]. In the United States, BPD affects 10,000–15,000 infants each year, with approximately 40% of VLBW infants receiving a diagnosis of BPD prior to discharge from the NICU [20]. As the treatment of premature infants has evolved, younger patients are surviving to discharge, and BPD is becoming more common both within the hospital and in the community.

For most infants, the clinical course of BPD is characterized by longitudinal respiratory support, punctuated by intermittent acute episodes requiring increased respiratory support. Although many babies with BPD can wean from their respiratory support to room air within the hospital, in other cases, a slow titration of oxygen may be completed at home. In cases of severe BPD, infants may require more advanced respiratory support, including home mechanical ventilation. Tracheostomy is required to facilitate long-term ventilation in these cases, which necessitates a surgical procedure, post-procedure ICU monitoring, prolonged ICU stays, and variable need for sedation and neuromuscular paralysis [25,26].

To provide the minimum requirements for home ventilation, a family requires access to a comprehensive medical home with accessible respiratory specialists, education on machine care, an attentive caregiver available at home at all times, and regular maintenance with replacements for equipment [27]. Currently, there are no standardized recommendations for weaning ventilation at home, however, this process is usually approached in a stepwise fashion over a period of months to years, providing the patient with initially short but increasingly longer periods of time off the ventilation machine with the eventual goal of tracheostomy decannulation. This time is not without dangers as baseline lung pathology and a long-term advanced airway leave the child at increased risk for more frequent and severe infections or even fatal tracheostomy emergencies, such as a mucus plug or accidental decannulation [28]. The course after initial tracheostomy surgery to home ventilation is often marked by difficulty with ventilation, high respiratory support needs, sedation needs, and muscular deconditioning. Future planning and development of expectations for home care are important to continue to move the care forward and eventually transition to outpatient care.

Even when well, patients require frequent clinical contacts to monitor clinical status and to evaluate the progress of the respiratory wean, and eventually proceed with surgical decannulation. Even after decannulation or freedom from respiratory support, they may continue to have sequelae due to residual lung disease or impacted nutrition and neurocognitive development [29,30]. With time and growth, most infants are able to tolerate the weaning of ventilator settings. Even most patients with severe lung disease are expected to achieve eventual freedom from mechanical ventilation with tracheostomy decannulation by the ages of 2.5–4 years of age [31]. Neurologic impairment makes the likelihood of a delay or inability to achieve this outcome more likely. Some patients will require lifelong mechanical ventilation, and it is important to prepare families for all these potential possibilities [32].

### 3.2. Palliative Care in Bronchopulmonary Dysplasia

The NICU is an inherently emotionally charged environment for families. In this setting, parents are often asked to make difficult decisions complicated by prognostic uncertainty. This is reflected in BPD, a clinical diagnosis characterized by chronic respiratory failure with a waxing and waning course and poor predictors for clinical outcomes. BPD patients are unique among those with chronic lung disease because their overall course follows clinical improvement over time. However, this progress is often obscured by day-to-day fluctuations in respiratory function and frequent setbacks due to complicating infections or other illnesses. These setbacks are helpful to proactively address as parents can easily become frustrated with the lack of forward progress. For families, this rollercoaster is not only devastating in the short term but both physically and emotionally exhausting over time [33]. To help combat this, forward progress, even in small steps, is important for families to see. Support of the parental burden by the primary medical teams in the NICU and post-discharge is important to the family’s well-being as a primary approach to palliative care. Specialized palliative care services may benefit these families early on as they grieve the loss of a prospect for a technology-free child in the short term. As each decision is made for more intensive support, there is often an accompanying cycle of fear, grief, and acceptance that the family endures. Other support during prolonged hospitalizations may include sibling support, memory making, and support of open communication between the parents and primary care teams.

Given the fluctuating clinical course of BPD, it is especially important to manage a family’s expectations early, preparing them both for long-term success and intermittent setbacks. In these cases, clarifying goals of care can help a treatment team to make unified decisions for the benefit of the patient. As the patient’s clinical course evolves and their long-term needs declare themselves, it is critical to educate families on how decisions not only affect the next steps in treatment but to prepare for home life. In addition, providers should be diligent in managing the multi-systemic sequelae of prematurity that often accompany severe BPD which can impair a patient’s well-being, including agitation, discomfort and pain, neurologic irritability, analgesic withdrawal, spasticity, reflux, dysphagia, and infection. As the child progresses, special attention should also be paid to monitoring developmental progress, participation in therapies, and swallow assessments. At the time to discharge from the hospital, there is a spectrum of support needs. Some infants are sent home with no respiratory support but may require enteral feeding or other therapies. Some are sent home with supplemental oxygen therapy, and some with the need for invasive mechanical ventilation. For patients requiring home oxygen, it is critical to educate families on home oxygen therapy, monitoring, and follow-up. All patients who are discharged on medical support devices also benefit from a thorough needs assessment as many families require supplemental financial support or may struggle through health literacy.

Evaluation for tracheostomy and home mechanical ventilation is warranted for patients who have chronic respiratory failure or chronic respiratory insufficiency. Discussion of tracheostomy is best handled as a multidisciplinary approach that not only explains the surgical procedure and risks but also examines life at home with ventilator care [25,26]. Given inherent unpredictability in the progression of BPD, for hospitalized infants with planned upcoming surgery, it is important to continually re-evaluate if tracheostomy remains the most prudent course of action. This includes reassessing further options for treatment and consent for tracheostomy as a patient’s clinical course evolves over the course of several days or weeks. A specialist with home ventilator management and palliative care team can partner to help families manage goals and expectations at home, transition to home with siblings, ensure home nursing availability, and provide an extra layer of support. Once the transition to outpatient care has been made this team can act as the palliative care team within the goals of care as they are established with the family. There may be some families who still benefit from outpatient specialized palliative care services, especially those with children who have multisystem sequelae of prematurity, or those who are not weaning as quickly as hoped.

## 4. Neuromuscular Disease

Nazar is an 18-year-old boy with Duchenne’s muscular dystrophy (DMD) who requires CPAP at night, has cardiomyopathy, and who is bed-bound from profound leg and trunk weakness. His current evaluation schedule includes outpatient PFTs and an echocardiogram every six months and has been told to abstain from caffeine. He would like to stop coming to the clinic so frequently because the tests only show disease progression which he finds to be depressing. He is afraid that his clinical trajectory is similar to his older brother, who died of DMD in his early 20s while on a home ventilator.

### 4.1. Neuromuscular Disease Clinical Course

Neuromuscular disorders (NMD) comprise a group of diseases that affect the motor neuron, neuromuscular junction, and muscle fibers leading to static, fluctuating, or progressive muscle weakness. They encompass a heterogeneous mix of inherited and acquired causes, including degenerative, metabolic, traumatic, immunologic, and toxic etiologies. Involvement of the inspiratory, expiratory, or bulbar muscle groups reduces an individual’s ability to cough effectively or take deep breaths. As a result, individuals may suffer from aspiration, recurrent respiratory infections, airway obstruction, sleep-disordered ventilation, and respiratory failure [34,35,36].

Just as NMDs vary in pathology, they vary in prognosis. A child diagnosed with Type 1 spinal muscular atrophy is highly likely to experience rapidly progressive respiratory failure during infancy [37], whereas a child diagnosed with Duchenne’s Muscular Dystrophy (DMD) is unlikely to suffer from respiratory failure until their second decade of life [38]. In still other cases, an exact etiology for the neuromuscular disorder may not be identified due to either unidentified genetic mutations or limitations in local diagnostic resources.

In cases of severe neuromuscular disease such as that seen in DMD, patients often experience non-linear progression in muscle weakness and respiratory insufficiency interrupted by intermittent severe illnesses which accelerate the onset of respiratory failure [39,40,41,42]. Although respiratory manifestations are often the primary drivers of symptoms towards the end of life, individuals with NMD often experience systemic complications including malnutrition, cardiac dysfunction [43], and scoliosis. In addition, patients often report significant musculoskeletal pain, fatigue, depressive mood, constipation, and dyspnea [44]. Despite the heterogeneous etiologies of these disorders, respiratory muscle training, mucociliary clearance, controlled ventilation, and nutritional support form the backbone of supportive care for patients with NMD. For patients with NMD affecting respiratory muscles, the introduction of non-invasive ventilation and eventual tracheostomy with home mechanical ventilation may be available as a life-prolonging measure [45]. However, in many cases, invasive long-term ventilation is not routinely recommended, but should instead be used in specific instances, such as immediately after the extubation of a child with SMA following an acute illness [46].

The past decade has seen substantial growth in the number of available novel therapies in treating specific neuromuscular disorders, especially Duchenne’s Muscular Dystrophy, Spinal Muscular Atrophy, and Limb Girdle Muscular Dystrophy. In broad terms, these treatments utilize a variety of approaches to either target the dysfunctional mutation through pre-mRNA splicing, replace mutated genes using a viral vector, or by upregulating other physiologic products in the muscle with similar properties to the dysfunctional protein [47,48]. For many patients, these therapies provide significant improvement in physical functioning, disease-related biomarkers, and slowing the progression of illness. However, many neuromuscular disorders do not yet have effective targeted therapies. Even within groups who are eligible to receive novel agents, accessibility remains a key barrier to patients receiving the medication [48].

### 4.2. Palliative Care in Neuromuscular Disease

The palliative care needs of individuals with Neuromuscular Disorders will vary greatly depending on their diagnosis and goal of care [49]. Prognostication may be offered to the patient and family at the time of diagnosis and with changes in condition. For example, boys with DMD currently live into their second, third, and rarely fourth decades. This generality can be shared at diagnosis by the primary physician or pulmonologist. However, this prognosis can become more specific over time and with the progression of the disease, such as after a hospitalization for pneumonia which can prompt discussions regarding life-sustaining care. Patients and families may desire very specific prognostic information such as length of life with or without certain treatments or interventions or information regarding symptom burden. These more detailed prognostic discussions can be led by a primary physician or subspecialist but may benefit from specialized palliative care support.

Patients and family members with DMD report uncertainty over the trajectory and management of NMD, particularly regarding the timing of discussions of end-of-life planning or how to connect to palliative care resources [44]. At the same time, parents report significant discomfort regarding discussions about death with their children with NMD. However, the same study showed a positive association between the frequency of these discussions and advanced care planning [44].

Advance care planning discussions and the completion of advance directives should be facilitated by the healthcare team. If the primary physician or subspecialist is not knowledgeable or comfortable in this area, specialized palliative care support or consultation should be pursued. These discussions should focus on the patient’s likely disease trajectory and involve specific consideration of life-sustaining or comfort-focused care a patient or family would desire such as an advanced airway, mechanical ventilation, and cardio-pulmonary resuscitation. Advance directives are for adult (18 and older) patients, but advance care planning is important for patients of any age to do with their families and healthcare teams. The degree to which pediatric patients want or can be involved in end-of-life decision-making and/or the designation of surrogate decision-makers is also important in advance care planning for patients with NMD. Advance care planning should take place at the time of diagnosis if appropriate and should be revisited at regular intervals (such as an annual appointment) and upon changes in condition. This regularity helps relieve the stress and anxiety that may be associated with advance care planning. Advanced care planning is best done when the patient is relatively healthy, and it is an iterative and dynamic process.

Palliation of symptoms in NMD may involve assessment and management of dyspnea, secretions, anxiety, and pain. In addition to providing symptomatic relief, all children, and families with complex medical conditions and with ESLD could benefit from social work, chaplaincy, and bereavement support. However, there are some unique psychosocial aspects of this disease process that may require intensive interdisciplinary support. In situations when patients experience anxiety and existential distress about the dying process, extra chaplaincy support and referrals to psychiatry and psychology can be helpful.

## 5. Pulmonary Hypertension

Vesa is a previously healthy six-year-old girl who presented to the Emergency Room (ER) after a syncopal event while playing outside with her siblings. Her family had noticed progressive exercise intolerance with shortness of breath for the past three to four months. In the ER, she is diagnosed with severe pulmonary hypertension requiring ICU admission and respiratory support, as well as initiation of multiple PH-directed therapies and vasoactive medications. Currently, she is being considered for extra-corporeal life support. Vesa’s parents have never heard of the diagnosis of pulmonary hypertension and are struggling to keep up with the rapid evolution of their daughter’s disease. They want the medical team to do whatever is necessary to save her life but are concerned that they are not making the right decisions for their daughter given the significant risks associated with each proposed intervention.

### 5.1. Pulmonary Hypertension Clinical Course

Pulmonary arterial hypertension (PAH) is a disease caused by the restriction of blood flow through the pulmonary arterial circulation that leads to increased pulmonary vascular resistance and ultimately right heart failure [50]. The diagnosis refers to a distinct subset of Pulmonary Hypertension patients (mean pulmonary artery pressure >20 mmHg) that comprises a diverse set of hereditary and de novo etiologies, including familial disease, connective tissue disease, congenital heart disease, portal hypertension, and idiopathic causes [51,52]. The clinical course of PAH typically consists of a long-term functional decline interrupted by acute deteriorations in clinical status due to right heart failure. As PAH progresses, patients most frequently describe worsening dyspnea on exertion, fatigue, drowsiness, and chest pain as daily symptoms which impede their activity [53]. In order to better describe the impact of PAH on each individual, the World Health Organization has further delineated a series of functional classes ranging from I (no limitation to physical activity) to IV (inability to carry out any activity without symptoms) that serve as an important predictor of prognosis, and which correlates with health-related quality of life [54].

Prior to disease-modifying therapies, the life expectancy of an individual newly diagnosed with PAH averaged 2.8 years from the initial diagnosis [55]. Despite interval advancements in both pharmaceutical and surgical interventions, PAH remains a progressive, symptomatic, life-limiting disease with poor survival outcomes past five years [56]. Treatment regimens are tailored to each patient, focusing on dilating and offloading the pressure on the pulmonary arterial circulation through the use of PH-directed therapies such as phosphodiesterase (PDE) 5 inhibitors, endothelin receptor antagonists, continuous prostacyclin infusions, palliative shunt construction, and lung transplant. Even with these interventions, patients with PAH are expected to worsen and progress to right ventricular failure over time.

Depending on the age and severity of the disease, PAH patients may be limited in activities compared to peers as exertion may exacerbate symptoms or even be life-threatening. Death may be predictable as right ventricular (RV) function worsens over time along with decreased activity tolerance. Because children often push through significant right ventricular strain, they may not appear as tenuous as they are. Even seemingly routine childhood illnesses may lead to a life-threatening exacerbation of pulmonary hypertension. This may lead to instances of sudden death, which are unexpected for both the family and their support team. Given PAH’s significant functional impact, the burden carried by caregivers, and the uncertainty associated with disease progression, patients and their families are at particularly high risk for anxiety, depression, and stress with several studies suggesting these comorbidities are undertreated in this population [57,58,59].

### 5.2. Palliative Care in Pulmonary Hypertension

Pulmonary Arterial Hypertension is a life-shortening and life-threatening disease with a potential for rapid clinical decompensation, making these patients ideal candidates for early specialized palliative care team involvement at the time of diagnosis. Regardless of specialized palliative care availability, providers should engage in vigilant primary palliation both surrounding the time of initial diagnosis and throughout the treatment course of these high-risk patients [60]. Patients with PAH and their families can specifically benefit from assistance with grief or loss associated with the diagnosis, advanced care planning, and palliation of treatment side effects.

In children with no prior family or personal history of PAH or lung disease, the diagnosis often arises in a previously healthy child. Grief surrounding a new diagnosis is often multifold, including the loss of a child, guilt over the hereditary component of the disease, and the potential implications for other family members moving forward. Providing patients and caregivers with a strong support system of counseling, education, and peer connections can help support a family through this particularly difficult initial period. Following the initial diagnosis, specialized palliative care teams can continue to provide in managing significant anxiety, depression, and existential distress that often accompanies the progression of PAH’s physical symptoms [57]. In addition, several factors aggravate the social impact of PAH: the disease is uncommon; it is not often associated with the external sign of illness; it reduces the ability to work or engage socially; it carries significant caregiver burden [57]. This highlights the importance of providing individualized education for patients and their families to help delineate expectations for clinical progress and to set a foundation for discussions of advanced care planning. Because of the high burden presented by physical and psychosocial stressors, it is appropriate to screen both patients and their caregivers for anxiety and depression at least annually and with significant changes in clinical status.

Given the poor outcomes and significant negative impact on HRQOL associated with the treatment of PAH, it is especially important to ensure patients and families understand their options for care early in a patient’s management course and set patient goals to guide further management [61]. Medication side effects, the methods of their administration, the need for further follow-up, and their clear limitations should be discussed prior to initiation. For many patients, the loss of fertility associated with the initiation of endothelin receptor antagonists is particularly difficult to handle. Patients on continuous intravenous prostanoid therapies to treat PAH also face unique challenges in respect to palliative and end-of-life care. In some health systems, hospice care and prostanoid therapy cannot be pursued simultaneously, yet patients and family members are often reluctant to decrease or discontinue prostacyclin, viewing this as intentionally hastening death [62]. These discussions should be addressed prior to the initiation of medical therapy to help mitigate potential conflicts between a patient’s goals of care and their management course. Along with medical therapies, invasive surgical measures may be offered for symptom reduction and survival benefit, including palliative shunts, pulmonary artery denervation, RV assist devices, and lung transplantation [63]. These options should be discussed early in a patient’s treatment course as the progression of the disease may preclude these procedures at a later time.

Symptom management presents unique challenges in PAH which require close coordination between Primary and specialized palliative care teams. Side effects of PDE 5 inhibitors, endothelin receptor antagonists, and continuous prostacyclin infusion provide ready targets for palliative management [64]. Along with encouraging early enrollment in pulmonary rehabilitation to assist with controlling breathlessness, patients may benefit from supplemental oxygen, opioids, and anxiolytics to manage dyspnea and anxiety [65]. However, care teams must take into account the potential hemodynamic effects of these interventions as any intervention that depresses systolic blood pressure, affects RV contractility, or increases RV afterload may lead to rapid worsening hypotension and death [57].

It is important to recognize that in some cases, a family’s choice to pursue comfort care measures through hospice and to forgo invasive or aggressive measures can be a loving decision in approaching a child’s care. This option should be revisited throughout a patient’s progression, especially prior to invasive measures such as the initiation of continuous prostanoid infusion or surgical intervention. Given the unpredictable nature of PAH, setting advanced care plans early in the clinical course is particularly important in the event of acute clinical decline. Facilitation of advanced care planning early in the disease can allow patients peace of mind to know priorities will be honored if they come to a stage when they can no longer speak for themselves and can help reduce some of the burdens of end-of-life decisions for next of kin [66].

## 6. Lung Transplant

Karolina is a 15-year-old girl who received a lung transplant six years ago for idiopathic pulmonary hypertension. Early in her post-transplant course, she was able to return to competitive swimming. However, her lung function has declined over the past 18 months, and she now becomes dyspneic at rest despite 2 L per minute of oxygen via nasal cannula. Karolina hopes for a second lung transplant but admits to non-adherence with her immunosuppressive medication. Her parents remember her prior wait on the transplant list and are worried that her function is declining too quickly for a repeat transplant.

### 6.1. Lung Transplant Clinical Course

When compared to other solid organ transplantation, the field of pediatric lung transplantation is relatively new. Since the first pediatric lung transplant was performed in the 1980s, the practice has now been expanded, with more than 100 pediatric lung transplants performed annually, with 90% of pediatric lung transplants occurring within the United States or Europe [67,68]. Pediatric lung transplants are exclusively pursued in the setting of end-stage lung disease, with the most common indications varying by age, including pulmonary hypertension in children under 5 years of age and cystic fibrosis in children older than 5 years of age [69]. Although pulmonary failure due to cystic fibrosis accounted for over 60% of lung transplants in the past two decades, the proportion of transplants required by cystic fibrosis patients is expected to continue declining in the near future given the success of evolving therapies for this disease

In the United States, eligibility for lung transplantation is determined by a Transplant Center. Once listed, the Lung Allocation Score (LAS) is calculated to reflect both the severity of the child’s illness and their potential for post-transplant success [70]. Although the LAS has improved the prioritization of donor lung allocation, the process from initial evaluation for a lung transplant to receiving a suitable lung can take several months. In the interim, patients require management for continued progression of their lung disease with concurrent optimization for surgery. In individual cases, a left ventricular assist device (LVAD), extra-corporeal membrane oxygenation (ECMO), or mechanical ventilation may be used in an attempt to improve survival to transplantation. Even so, 1.5–31% of children die while waiting for a transplant, with an associated higher risk occurring at lower-volume or adult transplant centers [71].

Inpatient recovery following a lung transplant may require as few as 10 days or up to several weeks of hospitalized care, depending on the individual patient, their comorbidities, and potential complications related to their care. Children require lifelong immunosuppression after the transplant, achieved through maintenance regimens that often include calcineurin inhibitors, antiproliferative agents, and corticosteroids [72]. Side effects of these medications include hypertension, diabetes, renal dysfunction, and neurologic complications such as seizures [73]. When successful, transplantation can provide several years of improved pulmonary function and engagement in daily life that would not have been possible with the child’s native lungs. Maintaining optimal lung function requires strict adherence to medications, close family emotional and financial support, and careful monitoring for signs of organ rejection. Unfortunately, patients experience a functional decline due to bronchiolitis obliterans, rejection, malignancy, or infection even with perfect adherence. The median survival remains slightly less than six years [69].

### 6.2. Palliative Care in Lung Transplant

Although lung transplantation provides many patients with a respite from end-stage lung disease, even successful management is followed by expected progressive functional decline and poor long-term survival outcomes. The choice to pursue lung transplantation provides important opportunities for both primary and specialized palliative care. A palliative approach is important in helping families to decide whether a lung transplant is the right choice for their child, and in providing symptom management with emotional support before and after the procedure.

The first step in any lung transplant evaluation should be determining a patient’s goals of care. Although often overlooked [74,75], an SPC team can play an invaluable role in helping to frame transplantation appropriately, as a tool to reach a patient’s goals rather than a goal in itself. Frank discussions should be conducted that outline the expected course for transplantation, including its significant long-term limitations, expected progressive pulmonary decline, required lifelong immunosuppression, potential complications, and potential for death while waiting for a match. Other options, including hospice care and continued medical therapy, should also be discussed as viable alternatives. Finally, active listening should be employed to ensure a family not only understands their options but also understands how each option may best address their loved one’s goals of care. Should a family choose to move forward with transplantation, this early involvement of the SPC team can serve as a source of support and symptom management in the time leading up to and following the transplant [74].

Given the severity of illness required for lung transplantation consideration, patients are often at high risk for symptomatic progression and clinical decline while awaiting lung transplantation. Contrary to previously reported physician concerns, SPC involvement during this period has been shown not to negatively impact eligibility for lung transplantation or outcomes following transplant [76]. Primary pulmonology and SPC teams can work together to treat the dyspnea, anxiety, pain, and malnutrition in patients with end-stage lung disease. This assistance serves to optimize patients for transplantation, while providing additional stability in the patient’s care team should clinical developments preclude the patient from receiving a transplant. The period waiting for transplantation provides unique psychological stressors on patients and their families. Availability of donor organs is unpredictable, yet expeditious transport and implantation is paramount to a transplant’s outcome. As a result, potential candidates and families must be prepared to report to the hospital at any time if they are notified of potential organs. This state of constant readiness can provide significant psychological strain on families, exacerbated by the disappointment of potential “false alarms” in which the family is called into the hospital for a potential transplant that ultimately does not work out. Awareness of this stress creates an opportunity for both primary teams and SPC to provide needed emotional reinforcement, connect with peer support, and screen for caregiver anxiety and depression.

Although the majority of consults for palliative care in transplant recipients are for symptom management, adult recipients also identified difficulty with physical activity, spiritual needs, and fear of impending death as major unmet needs that benefit from palliative care involvement [77,78]. Families also require significant psychosocial support due to caregiver fatigue attributed to complex medication regimens, complications from the transplant, and a high frequency of medical visits [79]. Although SPC expertise is typically utilized most extensively during the first year after transplant [78], primary teams should have a low threshold to re-connect patients with SPC as the patient’s clinical function begins to decline.

## 7. Palliative Care Perspectives in Guadalajara

Silvio is an eight-year-old boy who was first diagnosed with cystic fibrosis at three years of age. He lives in a rural town 45 miles from the hospital where he receives treatment, however, his family does not have their own transportation. Silvio’s family also does not have publicly-funded medical care, and they pay for his medical expenses independently. This limits their access to standard therapies including antibiotics, systemic medications, and airway clearance devices. He is currently hospitalized after three months of recurrent episodes of bronchopneumonia, progressive functional decline, and a new diagnosis of heart failure in the setting of severe pulmonary hypertension. The Specialized Palliative Care Team has been consulted to assist with modalities of non-invasive ventilation, to help ease Silvio’s symptomatic discomfort both in the hospital and at home, and to guide Silvio’s family in establishing goals of care.

Marcelo is a 10-year-old boy who was first diagnosed with cystic fibrosis at seven years of age, despite multiple prior hospitalizations. His medical expenses are supported by his family as they do not have publicly-funded medical care. Since his initial colonization of *Pseudomonas aeruginosa* at eight years old, he has had frequent pulmonary exacerbations despite multiple treatments with inhaled tobramycin and broad-spectrum IV antibiotics. At nine years old, his spirometry showed an FEV1 of 31%. He was not eligible for lung transplantation. At that point, the Specialized Palliative Care Team was consulted to facilitate a discussion of goals of care and to provide continuity of support in both the inpatient and outpatient setting. Currently, Marcelo has been readmitted to the hospital with worsening functional status and advancing bronchiectasis on CT. He and his family affirm their prior decision to decline invasive endotracheal intubation or cardiopulmonary resuscitation in the event of clinical decompensation and opt to pursue comfort-care measures. Due to the cost burden and the family stress regarding home end-of-life care, the family opts to continue Marcelo’s care in the inpatient setting. The Palliative Care Team assisted with symptomatic control, including subcutaneous morphine and midazolam, while facilitating his connections with loved ones and providing support to his family.

In low-income countries, access to molecular testing is limited by its high cost for individuals who do not have publicly-funded medical care. In this setting, we face scenarios of late cystic fibrosis diagnoses that are established by clinical manifestations and complementary testing. Patients frequently present with recurrent pulmonary exacerbations, malnutrition, and early persistent bacterial colonization associated with repeated hospitalizations. Many of the patients in our environment are candidates for lung transplantation much earlier than in countries that have specialized treatment centers [80]. In Mexico, lung transplantation is not an option available in any accessible way. Multiple factors shorten both the longevity and quality of life for patients, including the poor access to health services, the high cost of care carried by patients, and the lack of specialized care [81].

Providing palliative care in an early and individualized way facilitates better symptom control and improves the quality of life for both patients and their families. It allows the establishment of spiritual and emotional support, optimization in managing resources, and improves accessibility to medical care throughout the patient’s lives. A palliative care model also assists in creating advanced directives as contingencies for clinical deterioration and provides outpatient follow-up in complicated social situations, so its inclusion is best offered early in the patient’s treatment. However, this method of care is not consistently considered in this patient population.

In Guadalajara, availability of palliative care services is offered in the hospital model, however there is no hospice or equipment for home visits outside of the metropolitan area. As a result, the care we provide is remote, supported by a network of doctors close to the community, and relies on families trained to care for patients at home to provide symptom control. In this environment, we do not have access to home infusion pumps for analgesia or in situations where the use of oral medications is not possible. The family is trained and resources are provided for the use of devices, including subcutaneous medication pumps. The family is trained to perform pulmonary rehabilitation at home, and medical attention is available by the healthcare team 24/7 by phone.

Other entities that cause severe lung damage in pediatrics also require palliative care management, such as severe bronchopulmonary dysplasia, neuromuscular disorders, and severe pulmonary arterial hypertension. Although it is often impossible to predict the exact course of these conditions, the participation of the palliative care team helps to understand the trajectory of the diseases and thus establish the best goals in a realistic, individualized way for each patient. In advanced stages of diseases that cause severe lung damage, the cornerstone of treatment to improve quality of life is the control of symptoms, such as dyspnea, anxiety, chest pain. Therefore, treatment with opioids, anxiolytics, analgesics, and non-pharmacological measures is used to improve dyspnea, with total pain management as the focus.

Patients and their families require assistance from a multidisciplinary team because frequent exacerbations and recurrent hospitalizations are likely to occur and can lead the patient to the final stages of life. When complications of these diseases lead a patient to the terminal stages of life, accompaniment through the death process with medication to optimize analgesia and sedation becomes the priority of care. Having a specialized pediatric palliative care expert in the management of these patients provides access to comfort and dignity at the end of life. Palliative care teams can also provide emotional support for both families and their medical team, assist with the precision of therapeutic goals, and in balancing medical progress on an individualized basis for the patient and their family [82].

## 8. Conclusions

Regardless of the setting, the management of children with ESLD should be guided by the patient’s and family’s unique goals of care which are informed by their individual values, beliefs, and experiences. One family may choose to prioritize spending as much time with their child as possible by pursuing life-sustaining measures such as tracheostomy with home ventilation or lung transplantation. Another family may choose to focus on symptom management, avoiding potential sources of perceived suffering such as invasive procedures. For most families, their goals are not binary but lie somewhere on a dynamic spectrum between these two care options. The healthcare team should provide disease-specific information, including prognosis and treatment options when possible, and assist the patient and family in making healthcare choices that are consistent with their values and goals.

As we have reviewed, the effective management of children with ESLD requires a multidisciplinary approach that is both grounded in medical evidence and respectful of both the patient’s and their family’s story. Through the combined use of primary palliation, regular needs assessment, and specialized palliative care consultation, an interdisciplinary team can provide a comprehensive approach to reduce the suffering of both children with ESLD and their caregivers. Ultimately this approach seeks to guide families through the uncertainty of chronic illness, celebrating individual successes, providing support through inevitable difficulties, and ultimately helping patients reach their individualized goals.

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
