# Peer review of "Palliative Care in Pediatric Pulmonology"

_children, 2021, doi:10.3390/children8090802_

Round 1

Reviewer 1 Report

Excellent summary of the importance of palliative care in pediatric pulmonary patients

Author Response

Thank you for your kind words!

On the suggestion of other reviewers, we have focused our cases and worked to make them uniform in structure across the paper. We have left them open-ended to serve as an entry point for each topic that we hope helps to frame the discussion. We also hope this better ties our section on Palliative Care in Guadalajara into the paper as a cohesive unit, and better highlights the discussion section of that component of the paper (rather than getting lost in the cases).

Reviewer 2 Report

The Authors present a paper: " Palliative care in Pediatric Pulmonology" interesting, with a complete analysis and related to a worthy issue. I don't have any concern about the content of the paper but I have specific concern about the shape of the paper:

1. It is too long and in this way becomes boring

 2. there are too detailed reported information

I suggest the Authors to make this modification:

a) in the Introduction specify the meaning of Palliative Care (interventions not only at the end of life)

b) for the reported diseases (Cystic Fibrosis, Bronchopulmonary Dysplasia, Neuromuscolar disease,Pulmonary HypertensionLung Transplant) : the case report as introduction is fine BUT it should be condensed as well the history and the management in a box and the description of Palliative Care should be shortened.

c) the same should be done for point 7 . Palliative Care Prospective in Guadalajara

d) the Conclusion should better report the modality in Guadalajara to approach Palliative Care: multidisciplinary is fine BUT something summarizing the followed winning steps. The readers must learn their strategic approach having the possibility to adopt it in their Country.

Author Response

Thank you for your careful review! Your thoughtful comments are greatly appreciated.

a) We debated this topic extensively as a group. Because this article is appearing in a Palliative Care special edition, we were intentional not to go into detail on the definition of palliative care as the audience would be familiar with the principles of palliative care. However, we are happy to revisit that idea.

b) We condensed the cases and made them more uniform in presentation. The goal was for the cases to open or frame the ensuing discussion, and so they are left open-ended to do so. We rewrote them to hopefully better facilitate this goal.

c & d) We reconstructed the cases in section 7 extensively to make them uniform with the other cases. We hope this helps to better unify our section on Palliative Care in Guadalajara to the rest of the paper. This hopefully also helps to highlight the discussion section of these cases that was designed to address more specifics in the strategy of approaching palliative care in Guadalajara. This leaves Section 8 to serve as the overarching conclusion for the paper. However, if section 7 needs more specific intervention to better clarify your questions, we are happy to revisit it.

We hope that rewriting and formatting the cases helps to break up the paper into more approachable segments to help address your first point. We also tried to remove unnecessary components of the cases to help address your second point.
